# Variation in Genotype and DNA Methylation Patterns Based on Alcohol Use and CVD in the Korean Genome and Epidemiology Study (KoGES)

**DOI:** 10.3390/genes13020172

**Published:** 2022-01-19

**Authors:** Myoungjee Jung, Yeon-Soon Ahn, Sei-Jin Chang, Chun-Bae Kim, Kyoung Sook Jeong, Sang-Baek Koh, Jeong-An Gim

**Affiliations:** 1Department of Preventive Medicine, Yonsei University Wonju College of Medicine, 20 Ilsan-ro, Wonju 26426, Korea; laurie810305@naver.com (M.J.); chang0343@yonsei.ac.kr (S.-J.C.); kimcb@yonsei.ac.kr (C.-B.K.); 2Department of Preventive Medicine and Genomic Cohort Institute, Yonsei University Wonju College of Medicine, 20 Ilsan-ro, Wonju 26426, Korea; ysahn1203@gmail.com; 3Department of Occupational and Environmental Medicine, Yonsei University Wonju College of Medicine, 20 Ilsan-ro, Wonju 26426, Korea; bandyoem@naver.com; 4Institute of Genomic Cohort, Yonsei University Wonju College of Medicine, 20 Ilsan-ro, Wonju 26426, Korea; 5Medical Science Research Center, College of Medicine, Korea University, 8 Gamasan-ro 20-gil, Guro-gu, Seoul 08308, Korea

**Keywords:** alcohol consumption, chronic diseases, DNA methylation, genotype

## Abstract

Alcohol consumption can increase the risk of chronic diseases, such as myocardial infarction, coronary artery disease, hyperlipidemia, and hypertension. We aimed to assess the association between genotype, DNA methylation patterns, alcohol consumption, and chronic diseases in Korean population. We analyzed 8840 subjects for genotypes and 446 for DNA methylation among the 9351 subjects from the Korean Genome and Epidemiology Study (KoGES). We further divided both groups into two sub-groups according to the presence/absence of chronic diseases. We selected genes whose methylation varied significantly with alcohol consumption, and visualized genotype and DNA methylation patterns specific to each group. Genome-wide association study (GWAS) revealed single nucleotide polymorphisms (SNPs) *rs2074356* and *rs11066280* in HECT domain E3 ubiquitin protein ligase 4 (*HECTD4*) to be significantly associated with alcohol consumption in both the presence. The *rs12229654* genotype also displayed significantly different patterns with alcohol consumption. Furthermore, we retrieved differentially methylated regions (DMRs) from four groups based on sex and chronic diseases and compared them by drinking status. In genotype analysis, cardiovascular diseases (CVDs) showed a higher proportion in drinker than in non-drinker, but not in DMR analysis. Additionally, we analyzed the enriched Gene Ontology terms and Kyoto Gene and Genome Encyclopedia (KEGG) pathways and visualized the network, heatmap, and upset plot. We show that the pattern of DNA methylation associated with CVD is strongly influenced by alcoholism. Overall, this study identified genetic and epigenetic variants influenced by alcohol consumption and chronic diseases.

## 1. Introduction

Alcohol use disorder or alcoholism ranks sixth among factors that increase the risk of disease-related mortality and disability [1]. Moreover, alcoholism is a prevalent lifestyle factor relevant to chronic cardiovascular diseases (CVDs) such as myocardial infarction, coronary artery, hyperlipidemia, and hypertension [2]. The correlation between alcohol consumption and chronic diseases is well known, and restriction of alcohol intake is commonly recommended for the management of chronic diseases. To date, many studies have analyzed the association of alcohol-induced chronic diseases in preventive medicine and epidemiologic studies. However, a new approach based on omics data and data integration is required.

CVD is the leading cause of global morbidity and mortality in both men and women [3,4,5,6]. Monogenic conditions can also lead to severe premature CVD and early death, if unrecognized and untreated [7]. CVD also inflicts a significant economic burden on countries [6,8] as well individuals. Although alcohol-dependent people may have diverse medical conditions or diseases, the major cause of their death is CVD [9,10]. The association between alcohol intake and CVD is explained as the J-shaped curve. Since alcohol exerts complex effects on the cardiovascular system, elaborating the major pathways involved is challenging. Epidemiology-based, genetic, and epigenetic approaches have been employed to explain the relationship between alcoholism and cardiovascular diseases, and an integrated analysis is relevant.

A mendelian randomization study provides an alternative approach to establish the causal role of moderate alcohol use in a suitable population and reveals that a genetic variant affects alcohol metabolism and thereby alcohol use. Genetic variants of the several genes affect alcohol metabolism [11]. Cook and colleagues (2021) identified subgroups among Asian Americans at high risks of CVD-related conditions associated with excessive alcohol consumption—females of Chinese, Japanese, Korean, and Vietnamese ethnicities in higher *ALDH2*2* groups. Overall, members of the higher *ALDH2*2* ethnic groups were associated with lower risks of CVD-related conditions [12]. The genotype of the alcohol metabolism genes is related to alcohol-related CVD, and the relationship requires verification under other conditions.

DNA methylation, a major type of epigenetic modification, is associated with various complex human diseases and traits [13] and is potentially an important mechanism to regulate gene expression [14]. DNA methylation patterns are modulated by several diseases, including CVD [15]. DNA methylation affects the cytosine within CpG sites in the entire genome [7], and different diseases are associated with varying DNA methylation patterns. Previous studies have reported a genome-wide increase in the levels of DNA methylation in patients diagnosed with CVDs [16], as well as confirmed coronary artery disease (CAD) [17]. DNA methylation status is also affected by the environment [18], and the variation is associated with alcohol consumption [19]. Alcohol modulates the DNA methylation machinery [20] as well as the metabolism [21]. Hence, CpGs can serve as useful biomarkers for this disease, and the identification of a subset of CpGs involved in the disease can provide insights into the etiology of this disease [22]. In addition, identification of differentially methylated regions (DMRs) is necessary to understand the relationship between DNA methylation and its function in organisms. Epigenome-wide association studies (EWAS) have been performed to reveal the effects of DMRs on various diseases [21,22].

The genetic and behavioral differences between males and females need to be considered before analysis. From an epidemiological perspective, the prevalence and prognosis of chronic diseases based on large cohorts are sex dependent [23]. In addition, drinking habits differ according to sex. Epigenetically, female-specific hypermethylation is observed as a result of X-chromosome inactivation [24]. Therefore, in order to circumvent the sex bias, male and female populations should be analyzed separately. Alternatively, genes located on the sex chromosomes X and Y should be excluded before analysis. In this study, we separated the subjects into four groups based on the incidence of CVDs and sex.

So far, the associations have been studied between drinking and CVD, and the genotype has been reported to be correlated with alcoholism. However, very few studies have been undertaken to screen and analyze the genotypes and epi-genotypes in Koreans. In this study, we investigated the correlation between the occurrence of CVD and genotypes in alcoholic and non-alcoholic subjects. Our goals are to identify the SNPs associated with CVDs induced by alcohol. We also tried to find DMRs from four groups based on sex and CVDs by drinking status.

## 2. Materials and Methods

### 2.1. Study Subjects and Data Source

The epidemiological and genomic data sets of the present study were collected from a survey during 2001–2002 and were obtained from the Ansan and Ansung cohort of the Korean Genome and Epidemiology Study (KoGES) released by the Korea Center for Disease Control & Prevention [25]. Among the baseline participants (*n* = 10,030, Ansung: 5018; Ansan: 5012) of the Ansan and Ansung cohort, the DNA samples of 10,004 of the participants were available and genotyped. After the data were filtered using standard quality control procedures, the genotype data of 8840 participants were considered for further analysis [26].

Participants were 40–69 years of age and belonged to the Ansan and Ansung communities located in Gyeonggi province, South Korea. All participants signed an informed consent form. This study was approved by the Institutional Review Board (IRB) of Yonsei Wonju Severance Christian Hospital (Approval Number: CR320355) and Korea University (Approval Number: KUIRB-2020-0191-01) and performed in accordance with the Declaration of Helsinki.

### 2.2. Study Design

This study investigated the genetic and epigenetic risk factors of CVDs [27]. We divided the subjects into two groups based on the incidence of CVDs (myocardial infarction, coronary artery, hyperlipidemia, and hypertension) and an absence of CVDs. A flowchart of how subjects for genotype and methylation analyses were selected from 9351 participants is presented in Figure 1. As a case–control study, we compared drinkers and non-drinkers in the two groups, CVD and non-CVD groups. We then identified the frequently detected genotypes and epigenotypes via association studies.

Among 8468 people in the genotype analysis, 4579 were drinkers and 3889 were non-drinkers. A confusion matrix was developed for 1543 CVD subjects and 6925 non-CVD subjects (Figure 1). For the presence or absence of CVD, a genome-wide association study (GWAS) between drinkers and non-drinkers was performed.

To avoid sex-biased DNA methylation differences, we also divided the groups by two sexes. Of the 423 patients divided into the aforementioned groups, 202 were drinkers and 221 were non-drinkers. The female group with CVDs consisted of 45 individuals: nine drinkers and 36 non-drinkers. The female group with no CVDs consisted of 165 subjects: 42 drinkers and 123 non-drinkers. Among 30 males with CVDs, there were 15 drinkers and 15 non-drinkers (Figure 1). One hundred and eighty-three subjects with non-CVDs were identified among men, of which, 136 were classified as drinkers and 47 as non-drinkers. 

### 2.3. Definition and Measurement of Lifestyle Factors

The incidence of alcohol consumption and CVD were assessed using questionnaires designed by well-trained investigators. Questionnaires about alcohol consumption specifically investigated the presence or absence of alcohol, and the question was precisely framed as “Can’t you drink alcohol or don’t you drink from the beginning (for religious reasons, etc.)?” The subjects were required to respond in either “Yes, I do not drink alcohol” or “No (currently drinking)”.

CVD was considered as the dependent variable, and the CVDs were investigated in patients with myocardial infarction, coronary artery, hyperlipidemia, or hypertension. Questionnaires about the presence and absence of each condition were composed as “Have you ever been diagnosed with the following by a clinician?“, with the possible responses “Yes” or “No”. In all categorical questionnaire items, “Yes” was coded as 2 and “No” was coded as 1. If there was no answer, it was marked as a missing value, and “NA” was indicated when coding.

In the questionnaires of exercise intensity, the time spent per day was written by the subjects in minutes. Questions were separated by intensity, and the levels were stable, sedentary, light, moderate, and vigorous.

### 2.4. SNP Genotypes

The genotypes were obtained using Affymetrix Genome-Wide Human SNP array 5.0 (Affymetrix Inc., Santa Clara, CA, USA), which contains 500,568 SNPs constructed by the Korea Biobank Array. To examine the relationship between genetic polymorphism and CVD, we selected the significant SNPs by *p*-values.

### 2.5. DNA Methylation Analysis

In this study, genomic DNA extracted from the peripheral blood of the subjects was used for the evaluation of DNA methylation levels. Total 500 ng genomic DNA of each sample was modified by treatment with sodium bisulfite provided in the EZ DNA methylation kit (Zymo Research, Irvine, CA, USA), according to the manufacturer’s manual. Genome-wide DNA methylation was profiled using Illumina Infinium Human Methylation 450 k BeadChip (Illumina, San Diego, CA, USA), which contains over 485,000 CpG probes covering 99% of the RefSeq genes. Each CpG probe has a β value ranging between 0 and 1; higher methylation of CpG gives rise to a value closer to 1. We analyzed 364,050 CpG probes from 423 subjects, excluding the missing values Furthermore, the gene symbol and genomic location corresponding to each CpG probe of the Illumina 450k methylation chip were obtained using the “get Annotation” function of the IlluminaHumanMethylationEPICanno.ilm10b2.hg19 library provided by Bioconductor (https://www.bioconductor.org, accessed on 6 December 2021). 

### 2.6. Statistical Analysis of Genotypes and Methylation

Statistical analyses were performed using the PLINK software (version 1.9) (https://www.cog-genomics.org/plink2, accessed on 6 December 2021) and R software 4.0.5. The PLINK software was used to filter irrelevant SNPs, including those on the X and Y chromosomes and the mitochondrial genome. During PLINK analysis, genotyping call rate (0.05) was used as the SNP filtering parameter to remove missing genotypes of the SNPs [28]. Male and female subjects were analyzed in stratification. The association between SNPs and CVD disease related to alcohol consumption is represented by a Manhattan plot. The R software was used to generate the Manhattan plot using qqman package version 0.1.8 for genome-wide association studies (http://cran.r-project.org/web/packages/qqman/, accessed on 6 December 2021).

The DMRs were identified using a *t*-test performed between drinkers and non-drinkers in each of the four groups. The filtered DMRs were visualized as volcano plots and heatmaps. Enrichment analyses were performed using DMRs and visualized as KEGG and GO dot plots and gene-concept networks and heatmaps. To design the classification models, two machine learning methods, that is, decision trees and random forests were applied using the “rpart” and “randomForest” packages, respectively. Enrichment analysis was performed using the “pathfindR” package, and enrichment terms were subsequently retrieved as an upset plot.

## 3. Results

### 3.1. Study Processes

The process followed in this study for selection and classification of the study subjects has been represented as a flowchart in Figure 1. We recruited 9351 subjects. Five variables were used to distinguish between patients with CVDs and drinkers, and subjects with missing values were excluded. Therefore, 8953 subjects were finally followed. The GWAS was performed on 8468 subjects, which is the intersection of 8840 subjects whose genotype was analyzed and 8953 subjects with no missing values in the five variables. DNA methylation analysis was performed on 423 subjects, which is the intersection of 446 subjects whose methylation patterns was analyzed and 8953 subjects with no missing values in the five variables. A confusion matrix classifying drinkers and non-drinkers in the GWAS and methylation analyses is presented at the end of the flowchart.

### 3.2. Identification of Genotype Patterns between the Four Selected Conditions

The GWAS results between drinking status were summarized as top three most significant SNPs in two separated groups (Table 1) and visualized as two Manhattan plots (Figure 2). CVD had the lowest odds ratio (OR) of 0.2402 in *rs2074356* belonging to *HECTD4*, which was the most significant genotype. On the contrary, without CVD, *rs2074356* was also the most statistically significant genotype. Therefore, *HECTD4* showed statistically significant values in both classes. The top three genotypes were in the same order by OR (Table 1). In the Manhattan plot of the CVD group, a region with −log10 (*p*-value) > 8 was found on chromosome 12, and a region with log10 (*p*-value) > 5 was found on chromosome X (Figure 2, top). In the Manhattan plot of non-CVD group, a region with −log10 (*p*-value) > 8 was found on chromosomes 12 and X, and a region with log10 (*p*-value) > 5 was found on chromosome 4 (Figure 2, bottom). The two groups showed a similar pattern but a more significant statistical difference because more subjects were included in the non-CVD group.

### 3.3. DNA Methylation Analysis of Various Groups

We examined demographic variables to classify the subjects based on the incidence of CVD and drinking habits for methylation analysis, as shown in Table 2. Total 14 demographic and sociological variables, such as sex and age, associated with CVDs caused by drinking were analyzed. Among the remaining 423 participants, 75 (17.7%) were identified as CVD patients. Among female subjects, the 45 participants with CVD were classified into 9 drinkers (20.0%) and 36 non-drinkers. Among male subjects, the 30 participants with CVD were classified into 15 drinkers (50.0%) and 15 non-drinkers. Furthermore, among the 165 female participants with non-CVD, 42 were classified as drinkers (25.5%) and 123 as non-drinkers (74.5%). Among male subjects, 183 participants with non-CVD were classified as 136 drinkers (74.3%) and 47 non-drinkers.

The mean and standard deviation of age and BMI according to the drinking status in the CVD and non-CVD groups are presented. Exercising for more than 60 min per day by area, education, income, and intensity is presented. The most common chronic disease in the CVD group was high blood pressure, and coronary artery disease and hyperlipidemia were high in the drinkers group, but there was no statistically significant difference. Only one patient in the CVD group had myocardial infarction.

### 3.4. Identification of DMRs

Owing to the differences in DNA methylation patterns between female and male subjects, the groups were divided according to sex to avoid bias. Volcano plots (Figure 3) depict DNA methylation matrices of the four groups (classified by sexes and CVDs). Points that meet the conditions are marked in pink (highly methylated in drink conditions) or sky-blue (highly expressed in no-drink conditions). In each plot, the *X*-axis represents the fold-changes (FC) and the *Y*-axis represents the *p*-value on a −log_10_ scale. The area that satisfies the threshold of FC and *p*-value is represented with two vertical lines and one horizontal line. Based on these criteria, the differences in methylation levels were confirmed across the female, CVD and male, CVD groups (Figure 3A,B); and the female, non-CVD and male, non-CVD groups (Figure 3C,D). Four pairs of DMRs were finally screened from each group based on the methylation patterns and were represented as a heatmap (Figure 4). Heatmaps were generated for the female, CVD and male, CVD groups (Figure 4A,B) as well as the female, non-CVD and male, non-CVD groups (Figure 4C,D). The heatmap shows the variation in DNA methylation between the four groups. Column annotation bars indicate the two parameters in the samples; i.e., alcohol consumption (drink and no-drink), and sex (male and female). The pink and sky-blue bars indicate sex and drinking, respectively. The row annotation bars represent *p*-value and FC, and a clustering pattern is observed. In the FC bar, pink indicates a pattern indicating high methylation under drink conditions, whereas sky-blue indicates high methylation under non-drinking conditions. The PV bar indicates the *p*-value, and the intensity of gray bars indicates the significance of the difference between two groups.

### 3.5. Enrichment Analysis of DMRs

We obtained a list of results from the KEGG pathway and GO analyses for the four groups (Figure 5). The KEGG dot plots of female, CVD and male, CVD groups are represented in Figure 5A,B, while those of female, non-CVD and male, non-CVD groups are presented in Figure 5C,D. The KEGG pathways with a difference among four pairs of the four groups were significantly enriched by the DMRs. In the CVD group, the neuroactive ligand-receptor interaction was the most enriched in females and was statistically significant. In the same group, males were most enriched with multiple diseases associated with neurogeneration pathways, which was statistically significant. In the non-CVD group, the MAPK signaling pathway showed the common top gene ratio values in the two sexes. Pathway-related genes were enriched in the non-CVD group. Changes in the pathways of genes related to DMR between drinkers and non-drinkers were determined by the presence or absence of CVD rather than differences between the two sexes. Interestingly, disease-related pathways have been presented in the male and female CVD groups, and will be applied to study each disease, alcoholism, and CVD.

### 3.6. Network Analysis of DMRs

Based on the network analysis, gene-concept heatmaps were generated (Figure 6). In each heatmap, x and y axes indicate the genes and included terms, respectively. In addition, the highly expressed genes in the drinking group are indicated in red, while the genes showing low methylation are indicated in blue. In female groups, 17 and 11 terms were obtained, and their related genes were identified along with their methylation levels (Figure 6A,C). In male groups, only three terms were revealed in the gene-concept heatmaps (Figure 6B,D). Furthermore, a difference in terms was recorded between female and male groups. Two GO terms in male group were found to contain numerous genes (systemic arterial pressure and blood pressure finding).

### 3.7. Machine Learning Approaches for Analyzing DMRs

Random forest analysis was performed to describe classification criteria such as decision tree (Figure 7). Two plots were obtained from the random forest model. In the first plot, the number of trees grown for random forest models of four groups was ranked by permutation accuracy importance. In the plot, the x axis indicates the number of grown trees (*n* = 500), and the y axis represents the out-of-bag classification error. Four random forest models were designed, and the number of pre-training tree models was 500. The second plot was a variable importance plot. The top 30 genes retrieved after a random forest analysis were ranked as per the classification parameters. X axis indicates “mean decrease in Gini coefficients”. Gini means inequity, and Gini importance was calculated by the average gain of purity from the given genes. Similar to a high Gini importance, high- weighted genes indicate that the impurity at the decision node is lower at the bottom node.

In the 210 female subjects, the presence of drinking and CVD were used as factors to classify four groups by random forest. *CDH22*, *DPYSL5*, and *WSB2* were identified as the top three high weighted genes. In the random forest analysis, DMRs that classify four categories based on two variables were extracted. In 348 subjects belonging to the non-CVD group, five genes with a high mean decrease in the Gini coefficient were observed in the genes that suitably explained drinkers and the two sexes (*PSMD10*, *ERCC6L*, *MCART6*, *NKRF*, and *DGKK*).

## 4. Discussion

### 4.1. DNA Methylation in Male vs. Female

EWAS are a powerful way to understand the relationships between epigenetic variation and human diseases. In this study, we investigated the epigenetic differences associated with drinking status, and provided DMRs related to alcoholism in two sexes [29,30]. Differential methylation patterns of the two X chromosomes lead to distinction between the inactive X and active X chromosomes [30]. Since the males have one Y and one X chromosomes, the overall methylation patterns of the X chromosomes of both sexes are distinct. A few studies have attempted to compensate for these X chromosome differences [31,32]. In addition, besides sex chromosomes, thousands of CpG sites on autosomes also show dissimilar DNA methylation patterns between males and females [33,34]. Hence, the two sexes are considered an important covariate while undertaking methylation and phenotype association studies. In this study, we attempted to separate the two sexes during classification of the groups to eliminate the bias arising from using information from both sexes.

When extracting DMR from each of the four groups, similar numbers were observed when fold change and *p*-value criteria were similar. This can be visualized in Figure 3 and Figure 4, and the genes included in the DMR, presented in Figure 4, will be helpful for future research related to alcohol consumption and CVD. In the KEGG pathway enrichment analysis, terms with a similar pattern were enriched for both sexes of the CVD and non-CVD groups, respectively (Figure 5).

### 4.2. Alcohol Consumption and DNA Methylation

DNA methylation may play a role in the progression from normative to problematic drinking and is responsible for several adverse health outcomes associated with alcohol misuse [32,33]. Several studies have attempted EWAS of alcohol consumption by examining alcohol-related DNA methylation patterns among young adults at a heightened risk of alcohol use and associated health problems [19,34,35]. Previous research also showed that the process of alcoholic cirrhosis may directly influence DNA methylation [36]. Furthermore, alcohol cirrhosis has been associated with a multitude of comorbidities, including cardiovascular disease [37].

We divided two groups according to the presence or absence of CVDs, and then obtained genotypes and DMRs that were significantly different according to alcohol consumption in each group. By dividing each group according to sex when identifying DMR, sex-specific bias was removed and each sex-specific pattern was presented. The enriched KEGG pathways and gene-concept heatmap show that the presence of CVD reflects the enriched terms better than the sex (Figure 4 and Figure 6). In four random forest analyses, it was confirmed that the four groups could be classified with similar grades (Gini coefficient) using DMRs (Figure 7). A high mean decrease in the Gini coefficient was observed in the non-CVD group, because the non-CVD group had the highest number of subjects, that is, 348. Although there were 75 subjects with CVD, DMRs were presented to explain male and female, and drinker status. Among them, *TIMM17B* showed the highest Gini coefficient. For genes such as *CDH22*, *DPYSL5*, *WSB2*, *SNORA80*, *GDK7*, and *KCNK2*, which are located in the top of the list, follow-up studies related to alcohol use and CVD are likely necessary.

Therefore, if we combine our results with previous study results, the presence or absence of CVD will be an important relevant factor when studying DNA methylation for alcohol use.

### 4.3. SNP and DNA Methylation

The GWAS and EWAS conducted to date have discovered genotypes and DMRs that explain alcohol use and CVD. Techniques such as methylation quantitative trait loci (mQTL) for correlating SNPs and DMRs have been introduced and applied in several studies [38,39]. In particular, studies that integrate SNPs and DMRs have been conducted in many cases related to mental health. As it is difficult to explain mental health as a complex phenomenon with a single omics data, several integrated studies have been performed.

Although this study did not progress to mQTL, there was no commonality between the genotypes and DMRs discovered in our classification group. If the factors related to alcohol use and CVD in our study are identified in future studies and mQTL analysis is performed, a stronger explanation can be provided.

### 4.4. Limitations of This Study

A strength of our study was the discovery of genotypes and DMRs according to alcohol use and CVD in Koreans using various methods. However, our study has the following limitations, which should be further investigated.

First, when classifying drinkers and non-drinkers, former drinkers were included as non-drinkers. In a previous study, DMRs were classified into three groups, but statistically significant results were not obtained due to the limitation of the number of samples. Currently, the KoGES cohort is being expanded, and methylation analysis will be additionally performed. In the future, DMRs for the three groups will be extracted and compared with the results of this study.

Second, there were no common results between genotypes and DMRs, and mQTL analysis was not performed. This will be further analyzed and verified in future extended cohorts. Finally, there were more drinkers than non-drinkers in non-CVD males. In addition, drinkers showed relatively high education and income among males. There was a bias in cohort collection, and it is expected to be corrected with more cohorts in the future.

## 5. Conclusions

In summary, we show that genetic variation in HECTD4 is associated with CVD risk in the presence and absence of alcoholism and that alcoholism significantly influences the pattern of DMR methylation associated with CVD in DNA prepared from whole blood. DMRs from four groups based on sex and CVDs were identified and compared by drinking status. Statistical methods (*t*-test), KEGG enrichment analysis, and random forests were used to obtain DMRs for classification. In CVD, enriched terms were found to be related to neuron and MAPK pathway in non-CVD. This study is expected to provide important clues regarding the relationship of GWAS and EWAS with alcohol use and CVD.

## Figures and Tables

**Figure 1 genes-13-00172-f001:**
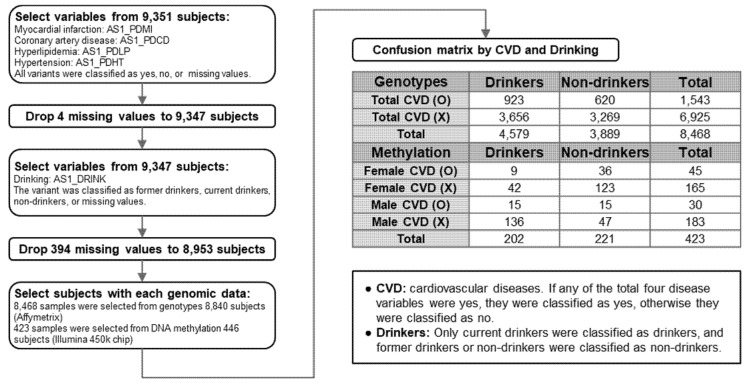
Flowchart for selection of the study population. The study population included 8840 genotypes and 446 methylation analysis results among 9351 subjects. Chronic cardiovascular diseases (CVDs) were considered as present when there was at least one of the following four chronic diseases: myocardial infarction, coronary artery, hyperlipidemia, and hypertension. “AS1_PDMI”, “AS1_PDCD”, “AS1_PDLP”, “AS1_PDHT”, and “AS1_DRINK” indicate the variable codes of myocardial infarction, coronary artery, hyperlipidemia, hypertension, and drinking status, respectively. Analysis was performed on 8468 subjects for genotyping and 423 subjects for methylation, which correspond to the intersection of the four aforementioned CVD-affected and drinking sample groups.

**Figure 2 genes-13-00172-f002:**
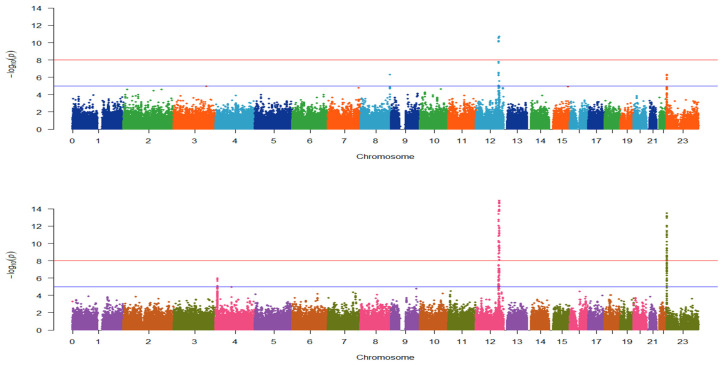
Manhattan plot of the association between drinkers and non-drinkers. (Top) Among 8468 subjects, 1543 subjects had at least chronic cardiovascular diseases (CVDs). In the CVD group, 923 were drinkers and 620 were non-drinkers. (Bottom) In the non-CVD group, 3656 were drinkers and 3269 were non-drinkers. The plot shows −log10 *p* value for each SNP against the chromosomal location.

**Figure 3 genes-13-00172-f003:**
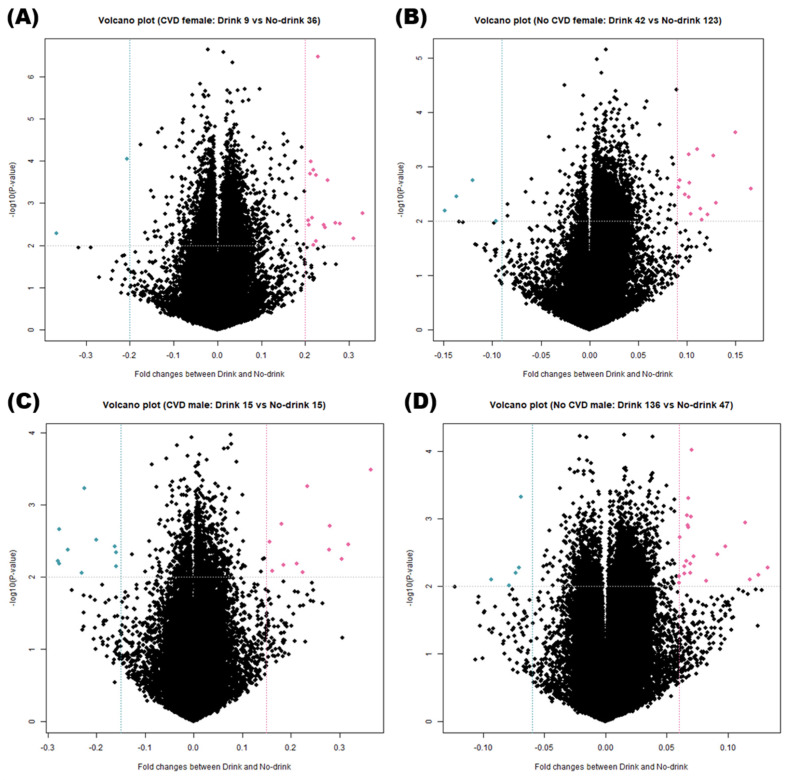
Volcano plot of differentially methylated regions (DMRs) indicating difference between four pairs of four groups. Total 423 subjects were divided into four groups based on the cardiovascular diseases (CVDs) (at least one of myocardial infarction, coronary artery, hyperlipidemia, and hypertension) or non-CVDs, and sex (male or female). (**A**) In females with CVD, *t*-test was performed on nine drinkers versus 36 non-drinkers, and 19 DMRs were found to satisfy the criteria |fold change| > 0.2 and *p*-value < 0.01. (**B**) In males with CVD, *t*-test was performed on 15 drinkers versus 15 non-drinkers, and 22 DMRs were found to satisfy the criteria |fold change| > 0.15 and *p*-value < 0.01. (**C**) In females with non-CVD, 42 drinkers versus 123 non-drinkers and 19 DMRs satisfied the criteria |fold change| > 0.09 and *p*-value < 0.01. (**D**) In females with CVD, the pairs of 136 drinkers versus 47 non-drinkers were subjected to *t*-test, and 27 DMRs satisfied the criteria |fold change| > 0.06 and *p*-value < 0.01. In the volcano plot, higher methylated regions between drinkers and non-drinkers of four groups are indicated as pink dots.

**Figure 4 genes-13-00172-f004:**
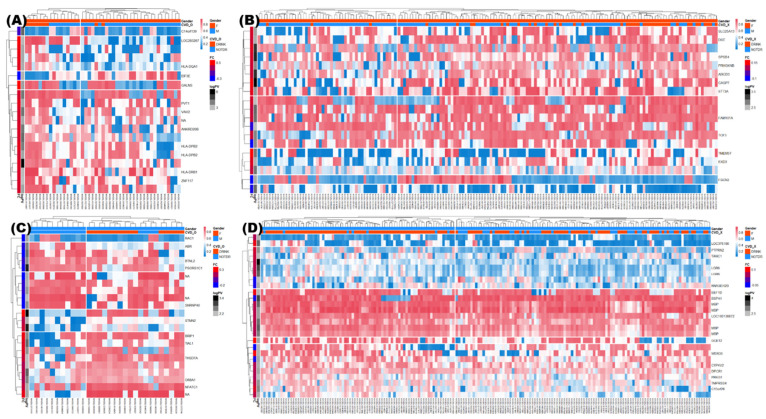
Heatmap of DMRs indicating the difference between four pairs of the four groups. Total 423 subjects were divided on the basis of presence of chronic cardiovascular diseases (CVDs); at least one of myocardial infarction, coronary artery, hyperlipidemia, and hypertension) or non-CVDs, and sexes (male or female). You may consider reporting this data in the body of results, rather than in the figure legends. This would help in avoiding the repetition between Figure 3 and Figure 4. (**A**) In females with CVD, *t*-test was performed on 9 drinkers versus 36 non-drinkers, and 19 DMRs were found to satisfy the criteria |fold change| > 0.2 and *p*-value < 0.01. (**B**) In males with CVD, *t*-test was performed on 15 drinkers versus 15 non-drinkers, and 22 DMRs were identified as satisfying the criteria |fold change| > 0.15 and *p*-value < 0.01. (**C**) In females with non-CVD, 42 drinkers versus 123 non-drinkers, and 19 DMRs were identified as satisfying the criteria |fold change| > 0.09 and *p*-value < 0.01. (**D**) In females with CVD, the pairs of 136 drinkers versus 47 non-drinkers were subjected to *t*-test, and 27 DMRs were identified as satisfying the criteria |fold change| > 0.06 and *p*-value < 0.01. In the heatmaps, higher methylated regions between drinkers and non-drinkers of the four groups are indicated as red annotation bars in drinkers. A higher significance deduced from a lower *p*-value is indicated as darker gray annotation bar. Sexes and CVD are indicated as column annotation bars.

**Figure 5 genes-13-00172-f005:**
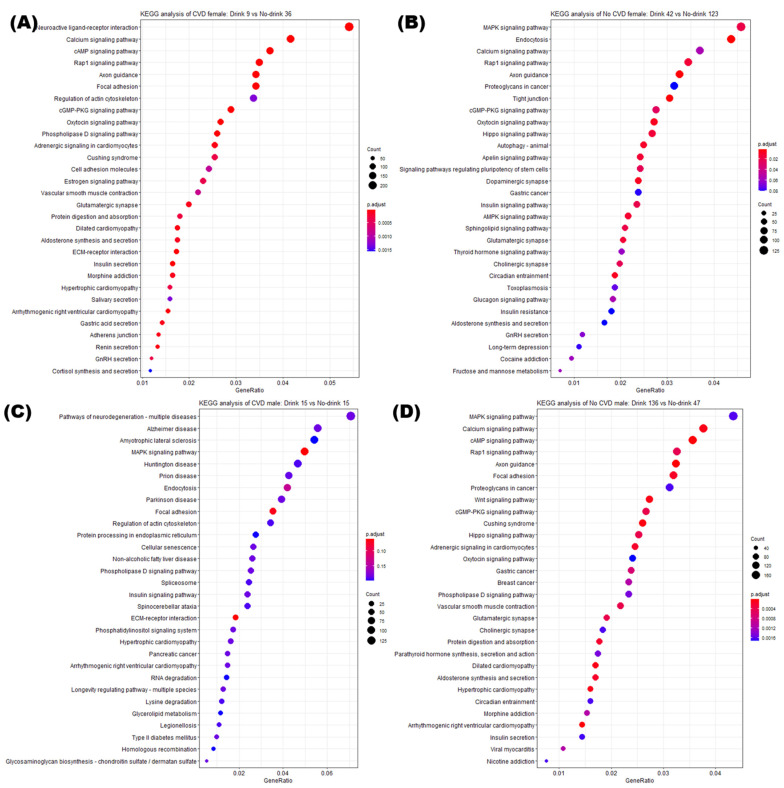
Enriched KEGG pathways of the DMRs indicating the difference between four pairs of the four groups. The top 30 enrichment KEGG pathways for DMRs satisfying the criteria *p*-value < 0.05 in each of the four pairs are presented. Color of the plot represents the *p*-value, and the size represents the count of genes. (**A**) In females with chronic cardiovascular diseases (CVDs), the pairs of 9 drinkers versus 36 non-drinkers were compared. (**B**) In males with CVD, the pairs of 15 drinkers versus 15 non-drinkers were compared. (**C**) In females with non-CVD, the pairs of 42 drinkers versus 123 non-drinkers were compared. (**D**) In females with CVD, the pairs of 136 drinkers versus 47 non-drinkers were compared.

**Figure 6 genes-13-00172-f006:**
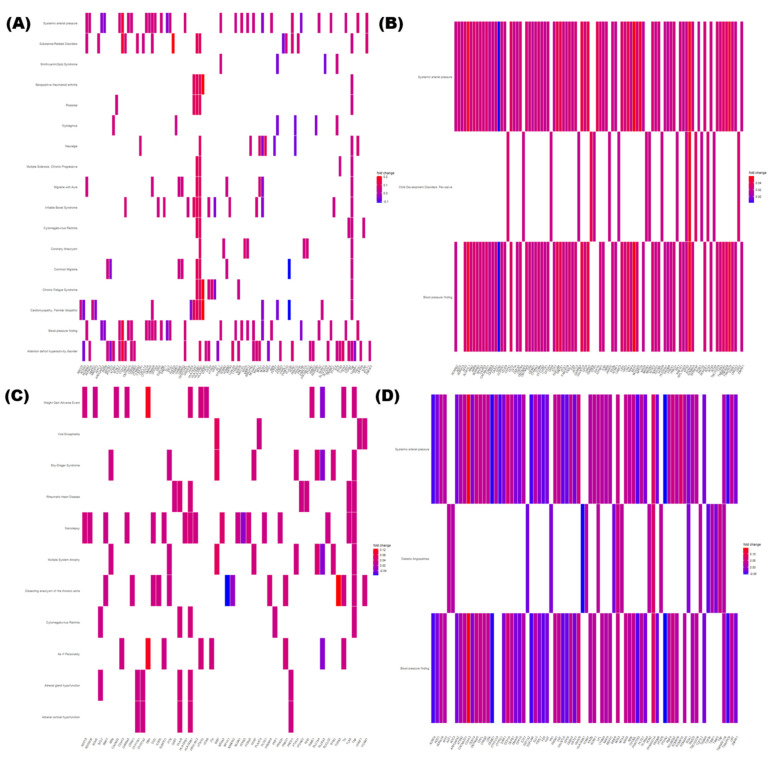
Gene-concept heatmap of the DMRs indicating the difference between four pairs of the four groups. The *X*-axis represents each gene, and the *Y*-axis represents the terms corresponding to each gene. Color ranges from red (high methylated in drinkers) to blue (high methylated in non-drinkers). (**A**) In females with chronic cardiovascular diseases (CVDs), the pairs of 9 drinkers versus 36 non-drinkers were included in the enrichment test, and each term satisfying the criteria |fold change| > 0.08 and *p*-value < 0.05 was retrieved. (**B**) In males with CVD, the pairs of 15 drinkers versus 15 non-drinkers were included in the enrichment test, and each term satisfying the criteria |fold change| > 0.08 and *p*-value < 0.05 was retrieved. (**C**) In females with non-CVD, the pairs of 42 drinkers versus 123 non-drinkers were subjected to the enrichment test, and each term satisfying the criteria |fold change| > 0.04 and *p*-value < 0.05 was retrieved. (**D**) In males with non-CVD, the pairs of 136 drinkers versus 47 non-drinkers were subjected to the enrichment test, and each term satisfying the criteria |fold change| > 0.04 and *p*-value < 0.05 was retrieved.

**Figure 7 genes-13-00172-f007:**
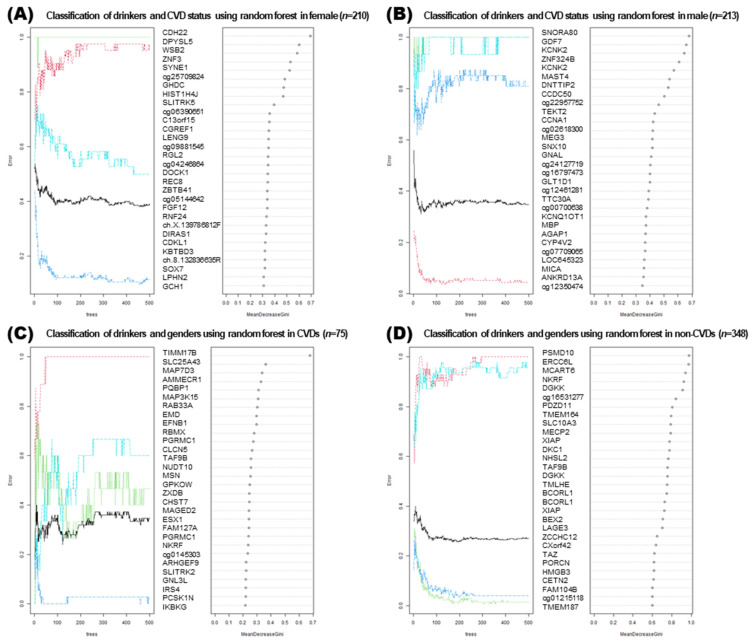
Random forest analysis results classifying the four groups from 423 samples. Each of the four random forest outputs was provided as an error plot and variable importance plot. Three conditions (sex, drinking, and chronic cardiovascular diseases; CVDs) were considered as criteria for classification. Subjects afflicted with at least one of the diseases, including myocardial infarction, coronary artery, hyperlipidemia, and hypertension, were included as subjects with CVDs. Variable importance plot of the random forest analysis resulted from the integration of a large number of models built by classification of each of the four groups. The variables are ordered top-to-bottom as most-to-least important in classifying the four groups. The ranked list of variables indicates the importance of each variable in classifying the data. The figure shows the top 30 variables important in the classification of the four groups. Four groups were divided based on the drinking and CVD status (**A**) from 210 female samples and (**B**) from 213 male samples. Similarly, four groups each were divided based on the drinking status and sex (**C**) from 75 CVD samples and (**D**) from 348 non-CVD samples.

**Table 1 genes-13-00172-t001:** The most significant SNPs associated with chronic cardiovascular diseases (CVDs) and without CVD between drinking status.

Class	Location (hg38)	dbSNP ID	Gene	Odds Ratio	*p*-Value
With CVD	chr12:112207597	*rs2074356*	*HECTD4*: Intron Variant	0.2402	1.49 × 10^−26^
With CVD	chr12:112379979	*rs11066280*	*HECTD4*: Intron Variant	0.2919	1.15 × 10^−24^
With CVD	chr12:110976657	*rs12229654*	None	0.308	2.01 × 10^−20^
Without CVD	chr12:112207597	*rs2074356*	*HECTD4*: Intron Variant	0.3013	1.41 × 10^−107^
Without CVD	chr12:112379979	*rs11066280*	*HECTD4*: Intron Variant	0.3326	1.95 × 10^−106^
Without CVD	chr12:110976657	*rs12229654*	None	0.3689	1.43 × 10^−75^

**Table 2 genes-13-00172-t002:** Basal characteristics of participants classified according to drinking and chronic cardiovascular diseases (CVDs) for methylation analysis.

Variable	CVDs (*n* = 75)			Non-CVDs (*n* = 348)		
	Never Drinkers	Drinkers	*p*-Value	Never Drinkers	Drinkers	*p*-Value
Sex	15/36/29.41	15/9/62.5	0.011	47/123/27.65	136/42/76.4	<0.001
Age	56.75 ± 7.63	53.62 ± 6.97	<0.001	53.12 ± 8.37	49.57 ± 7.95	<0.001
BMI	26.58 ± 3.05	25.5 ± 3.27	<0.001	23.91 ± 3.21	24.53 ± 3.39	<0.001
Area	34/17/66.67	14/10/58.33	0.607	102/68/60.00	72/106/40.45	0.001
Education	37/13/74.00	11/13/45.83	0.022	112/58/65.88	68/108/38.64	<0.001
Income	33/16/67.35	8/16/33.33	0.011	97/69/58.43	70/108/39.33	0.001
Exercise1	33/17/66.00	13/11/54.17	0.443	114/55/67.46	100/77/56.50	0.046
Exercise2	12/37/24.49	7/17/29.17	0.778	43/125/25.60	35/141/19.89	0.246
Exercise3	20/29/40.82	9/15/37.50	1.000	55/113/32.74	46/129/26.29	0.195
Exercise4	37/10/78.72	19/5/79.17	1.000	120/47/71.86	127/48/72.57	0.904
Exercise5	32/16/66.67	18/6/75	0.591	111/58/65.68	125/52/70.62	0.356
Myocardial infarction	1/50/1.96	0/24/0	<0.001	0/170/0	0/178/0	1.000
Coronary artery disease	3/48/5.88	3/21/12.50	0.377	0/170/0	0/178/0	1.000
Hyperlipidemia	3/48/5.88	4/20/16.67	0.201	0/170/0	0/178/0	1.000
High blood pressure	45/6/88.24	20/4/83.33	0.717	0/170/0	0/178/0	1.000

Continuous variables are indicated as “average ± standard deviation”, and categorical variables are indicated as “Counted samples in group 1/Counted samples in group 2/Ratio of group 1 in total samples”. Sex: Male = 1, Female = 2; Area: Ansung = 1, Ansan = 2; Education: Below middle school =1, Over high school = 2; Income: Under 1.5 million won =1, over 1.5 million won = 2; Exercise status: Under 60 min/day = 1, over 60 min/day = 2. Exercise levels are indicated as 1, 2, 3, 4, and 5 corresponding to stable, sedentary, light, moderate, and vigorous, respectively.; Four CVD diseases: disease group = 1, normal group = 2.

## Data Availability

KoGES dataset information and data shearing processes can be obtained from the site provided by the National Research Institute of Health, the Korea Disease Control and Prevention Agency, Ministry for Health and Welfare, Korea (https://nih.go.kr/contents.es?mid=a50401010100, accessed on 6 December 2021). R source codes will be provided upon request.

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
