# Peer review of "Variation in Genotype and DNA Methylation Patterns Based on Alcohol Use and CVD in the Korean Genome and Epidemiology Study (KoGES)"

_genes, 2022, doi:10.3390/genes13020172_

Round 1

Reviewer 1 Report

Dear Editor,

I was asked to review the manuscript with the following title:

"Variation in Genotype and DNA Methylation Patterns based on Alcohol Use and CVD in the Korean Genome and Epidemiology Study (KoGES)"

My overall evaluation of the manuscript is that the study design is well structured, but the results are not well organized or preseneted. Somehow there was no clear conclusion from each section of the  results and why the authors brought up some specific concepts e.g., KEGG pathways and GO terms, what are they for! No single explanation about any of the pathways or the temrs.

Specific comments:

1- 2.2 Study design: It is too complicated to understand, please provide a flow chart.

2- 3.2 Identification... : Why is it too short?

3- Figure 2: What does this figure tell, where does it belong to in the results, and why does the figure legend explain a different story as what exists in the figure?

4- 3.3 DNA... : Where is the table or the figure?

5- 3.4 Identification... : This is more of a methodological description.

6- Discussion: The discussion has a style of a review and does not tell any connection between the results.

Author Response

My overall evaluation of the manuscript is that the study design is well structured, but the results are not well organized or preseneted. Somehow there was no clear conclusion from each section of the  results and why the authors brought up some specific concepts e.g., KEGG pathways and GO terms, what are they for! No single explanation about any of the pathways or the temrs.

Response: We appreciate your valuable comment. We have thoroughly revised our manuscript accordingly.

  1. Defined “KEGG” [Lines 33-34] and “DMR” [Line 80]. An abbreviation has been added and deleted in Lines 42 and 49,
  2. Awkward paragraph division has been corrected [Lines 78-79], and additional explanations have been provided for unclear contents [Lines 79–83].
  3. The Introduction has been carefully rechecked, and appropriate studies have bene cited where necessary [Lines 83, 86, 88, and 105].
  4. Sections 2.3, 2.4, and 2.6 have been thoroughly revised.
  5. In Sections 3.5 and 3.7, the description of the results has been improved [Lines 305–311, 359–363], to clearly show our interpretation. We have discussed the results in the Discussion [Lines 390–395 and 405–421].

Specific comments:

1- 2.2 Study design: It is too complicated to understand, please provide a flow chart.

Response: We have thoroughly revised Section 2.2 and modified Figure 1. We have added information about GWAS [Lines 119–127]. In the confusion matrix added at the end of the flowchart, the number of subjects in each group is presented [Line 203]. Similarly, Section 3.1 has also been revised [Lines 193–202]. The Figure 1 legend has also been revised [Lines 207–208].

2- 3.2 Identification... : Why is it too short?

3- Figure 2: What does this figure tell, where does it belong to in the results, and why does the figure legend explain a different story as what exists in the figure?

Response: In the original version of our manuscript, Section 3.2 and Figure 2 were related to GWAS, and Sections 3.3 and 3.4 were the results of DNA methylation analysis. We have added the relevant descriptions in Section 3.2, and referred to Figure 2 [Lines 212–224].

4- 3.3 DNA... : Where is the table or the figure?

Response: We have changed the positions of Tables 1 and 2 and changed the layout of Table 2.

  1. In Table 2, there were errors related to the CVD parameters. When classifying values corresponding to each variable in Table 2, the “filter” function of the “dplyr” package and the “table” (a default function of R) were used. However, due to a logical error in the source code, the presence or absence of CVD was not accurately indicated. We have corrected the logical error of the R source code in Table 2 [Lines 239–246].
  2. We have added a description of Table 2 in Section 3.3 [Lines 244–245 and 258–263].

5- 3.4 Identification... : This is more of a methodological description.

Response: We agree with your comment. We have presented Section 3.4 as Section 2.6. The title of Section 2.6 has been changed and the relevant contents have been added [Lines 172 and 182–189].

6- Discussion: The discussion has a style of a review and does not tell any connection between the results.

Response: We agree with your comment. We have added the necessary information in the Discussion.

  1. Among the three sections, one section has been deleted because the section was not related to the study and was like a review.
  2. In the remaining two sections, we have added descriptions related to the results of this study [Lines 403–408 and 418–472]. Based on the correlation between the results shown in Figures 4, 6, and 7 and the preceding results, we have concluded that a follow-up study is necessary.
  3. Two new sections have been added, one for DNA methylation and SNP analysis (Section 4.3) [Lines 436–447] and the other for the limitations of this study (Section 4.4) [Lines 449–464].
  4. The Conclusion has been revised for clarity [Lines 465–472].

Reviewer 2 Report

Myoungjee Jung presented a manuscript titled „ Variation in Genotype and DNA Methylation Patterns based on Alcohol Use and CVD in the Korean Genome and Epidemiology Study (KoGES)”. Even though the topic is interesting the manuscript itself has some major flaws and ambiguities that need to be addressed.

  1. The manuscript has some serious English spelling and grammar flaws. Please thoroughly revise the whole manuscript.
  2. “Not all CVDs occur as a consequence of alcoholism”. I believe this is well-established so please expand this with a meaning or remove it.
  3. Expand the last part of your Introduction section with a brief paragraph about the aims and goals of your study.
  4. Moreover, your last paragraph where you elaborate why did you decide to do this study is insufficient. Merely stating that only few studies about the genotypes in Korean people is not enough. Please give a more elaborated explanation why did you choose to investigate the association between CVDs, alcohol consumption and genetic traits.
  5. Why did you include only subjects of 40-69 years of age? Since this is a epidemiologic study shouldn’t you include all adults?
  6. It is not clear how did you choose the included 423 patients. Were they chosen from the 8840 from the previous cohort? Please, you need to explain this clearly in your Methods section.
  7. Subsection 2.3. has some strange typo in the first paragraph.
  8. You need to explain with more detail the questionnaire used for lifestyle factors.
  9. When was the blood samples extracted? Your methodology regarding the included subjects is unclear, hence it is hard to understand everything afterwards.
  10. Figure 1 and the following descriptive text should be moved to the Methods section. Moreover, the flowchart is very incomprehensible and needs to be revised. Please add all the abbreviations to the legend.
  11. Moreover, you stated in Figure 1 that CVDs were considered when one of the following was present: myocardial infarction, coronary artery, hyperlipidemia, and hypertension. Firstly, everyone has coronary arteries so please revise this with the meaning you aimed for. Secondly, hyperlipidemia and hypertension are not CVDs in a narrower sense. They are major risk factors for developing CVDs but I would never categorize them as CVDs. This is a major weakness of your methodology.
  12. What statistical methods did you use to calculate the p-value in Table 1? Moreover, how did you estimate the normality of distribution? What methods did you use to normalize a variable that was not normally distributed?
  13. Qualitative variables in Table 1 are unclearly presented. Please revise that.
  14. Since your methodology is very weak, it is impossible to interpret your Results and comment your Discussion and Conclusion.

Author Response

Myoungjee Jung presented a manuscript titled „ Variation in Genotype and DNA Methylation Patterns based on Alcohol Use and CVD in the Korean Genome and Epidemiology Study (KoGES)”. Even though the topic is interesting the manuscript itself has some major flaws and ambiguities that need to be addressed.

Response: We appreciate your valuable comment. We have thoroughly revised our manuscript accordingly. Your comments helped a lot in the development of our manuscript.

  1. The manuscript has some serious English spelling and grammar flaws. Please thoroughly revise the whole manuscript.

Response: The revised manuscript has been edited and rechecked for language and clarity by Editage, a professional English language editing service [WYUCM_312_2]. Please find attached the relevant certificate.

  1. “Not all CVDs occur as a consequence of alcoholism”. I believe this is well-established so please expand this with a meaning or remove it.

Response: We agree the reviewer’s comment, and removed “Not all CVDs occur as a consequence of alcoholism” [Line 58].

  1. Expand the last part of your Introduction section with a brief paragraph about the aims and goals of your study.

Response: We added the goals of our studies as two sentences at the end of the introduction section [Lines 97-99].

  1. Moreover, your last paragraph where you elaborate why did you decide to do this study is insufficient. Merely stating that only few studies about the genotypes in Korean people is not enough. Please give a more elaborated explanation why did you choose to investigate the association between CVDs, alcohol consumption and genetic traits.

Response: We compared genetic information and epigenetic information provided by KoGES according to alcohol consumption and summarized the analysis results as a conclusion. This clearly shows the goal presented at the end of the Introduction section. [Lines 465–472].

  1. Why did you include only subjects of 40-69 years of age? Since this is a epidemiologic study shouldn’t you include all adults?

Response: The cohort included in KoGES was only recruited from adults aged 40-69 years. Therefore, we had no choice but to study the given subjects [Kim, Y.; Han, B.G.; KoGES Group. Cohort profile: the Korean genome and epidemiology study (KoGES) consortium. Int J Epidemiol 2017 Apr, 46(2), e20].

  1. It is not clear how did you choose the included 423 patients. Were they chosen from the 8840 from the previous cohort? Please, you need to explain this clearly in your Methods section.

Response: We have thoroughly revised Section 2.2 and modified Figure 1. We have added information about GWAS [Lines 119–127]. In the confusion matrix added at the end of the flowchart, the number of subjects in each group is presented [Line 203]. Similarly, Section 3.1 has also been revised [Lines 193–202]. The Figure 1 legend has also been revised [Lines 207–208].

  1. Subsection 2.3. has some strange typo in the first paragraph.

Response: The comments inserted in the manuscript during the editing process have been deleted.

  1. You need to explain with more detail the questionnaire used for lifestyle factors.

Response: Irrelevant contents have been deleted, and explanations for variables and coding methods for exercise have bene added. Similarly, we have added explanations for variables in the legend of Figure 1 [Lines 147–152]. Then, we have added explanations regarding exercise intensity in Table 2 [Lines 239–246]. Table 1 has been replaced with Table 2.

  1. When was the blood samples extracted? Your methodology regarding the included subjects is unclear, hence it is hard to understand everything afterwards.

Response: We analyzed the results collected through KoGES, and details on cohort recruitment and specific sample collection methods are presented in Kim et al., 2017 paper. KoGES related explanations were added, and related papers were cited [Lines 104-105].

  1. Figure 1 and the following descriptive text should be moved to the Methods section. Moreover, the flowchart is very incomprehensible and needs to be revised. Please add all the abbreviations to the legend.

Response: We changed the layout of Figure 1 to make it easier to understand and suggested each variable and the code names of the variables [Lines 207-208]. A table is included at the end of the flowchart, and the number of subjects included in each category is presented. A description of the modified Figure 1 was added to section 3.1 [Lines 193-202].

  1. Moreover, you stated in Figure 1 that CVDs were considered when one of the following was present: myocardial infarction, coronary artery, hyperlipidemia, and hypertension. Firstly, everyone has coronary arteries so please revise this with the meaning you aimed for. Secondly, hyperlipidemia and hypertension are not CVDs in a narrower sense. They are major risk factors for developing CVDs but I would never categorize them as CVDs. This is a major weakness of your methodology.

Response: We also agree with the comments about the weaknesses of the methodologies of the judges. This cohort was not a variable diagnosed with CVD, but was based on the results of a questionnaire submitted by subjects. Therefore, although it is not an accurate CVD diagnosis result, it was sufficient to obtain statistical significance because a sufficiently large number of subjects (n=75) were included.

  1. What statistical methods did you use to calculate the p-value in Table 1? Moreover, how did you estimate the normality of distribution? What methods did you use to normalize a variable that was not normally distributed?
  2. Qualitative variables in Table 1 are unclearly presented. Please revise that.

Response: For the p-value in Table 2 (originally Table 1, the order was adjusted), “t.test”, the default R function, was used for each variable. Then, “p.value”, one of the output variables, was used. Normality in both groups was confirmed by histogram using R's default function "hist". Since a sufficient number of subjects was secured for each group, the analysis was conducted without any statistical problems. For categorical variables, R's default function "fisher.test" for the coufusion matrix was used, and the output "p.value" value was used. In Table 2, there were errors related to the CVD parameters. When classifying values corresponding to each variable in Table 2, the “filter” function of the “dplyr” package and the “table” (a default function of R) were used. However, due to a logical error in the source code, the presence or absence of CVD was not accurately indicated. We have corrected the logical error of the R source code in Table 2 [Lines 239–246].

  1. Since your methodology is very weak, it is impossible to interpret your Results and comment your Discussion and Conclusion.

Response: We have thoroughly revised Materials and Methods section. In all sections, necessary parts were added, and unclear parts were clearly corrected or deleted.

Reviewer 3 Report

This is a study on the assessment of the association among alcohol consumption, four cardiovascular diseases (CVDs), SNP genotype and DNA methylation patterns in the South Korean population. Authors performed the Genome-wide SNP array and the Genome-wide DNA methylation profiling with the Affymetrix and Illumina technology, respectively. Additionally for the differentially methylated regions (DMRs), they analyzed the enrichment terms and KEGG ontology, and visualized network, heatmap and upset plot.

Authors may get some interesting results, but they are not able to present them properly. Significant findings should be described in the Results and Discussion, or it should be mentioned that there was not found any significance.

Major comments:

  • It is not correct to include the former-drinkers into the non-drinkers.
  • You have found that, in the male group with non-CVDs (N=183), there is much more drinkers (N=136) than non-drinkers (N=47). Does it mean that the alcohol use in man is a certain protection against CVD?
  • According to the Table 1, drinkers are particularly men with higher education and higher income. Does it mean that the alcohol use is a stress compensation? Then, there is larger environment effect in men, and women have higher genetic predisposition. Can you see differences in genetic or epigenetic predisposition to drinking between male and female?
  • In the Discussion, you should include the chapter about SNP results and remove the chapter 4.1 that is redundant (basic information) and you should summarize your results of DNA methylation.

Minor comments:

  • In the text, you have nonsense sentences (line 29-30, 127-128, 134-140) that should be removed.
  • The chapter “Definition & Measurement of Lifestyle Factors” should be explained in more details and clearly.
  • In the Table 1, it is not explained meaning Exercise1, Exercise2, Exercise3, Exercise4, and Exercise5.
  • The chapter 3.3 is redundant.
  • There are many technical and grammar errors.

Author Response

This is a study on the assessment of the association among alcohol consumption, four cardiovascular diseases (CVDs), SNP genotype and DNA methylation patterns in the South Korean population. Authors performed the Genome-wide SNP array and the Genome-wide DNA methylation profiling with the Affymetrix and Illumina technology, respectively. Additionally for the differentially methylated regions (DMRs), they analyzed the enrichment terms and KEGG ontology, and visualized network, heatmap and upset plot.

Authors may get some interesting results, but they are not able to present them properly. Significant findings should be described in the Results and Discussion, or it should be mentioned that there was not found any significance.

Response: We appreciate your valuable comment. We have thoroughly revised our manuscript according to these suggestions. Unnecessary parts have been deleted from the Discussion; we have discussed our results in light of previous studies. In Sections 3.5 and 3.7, the description of the results has been improved [Lines 317–324 and 372–376], to clearly show our interpretation. We have discussed the results in the Discussion [Lines 403–408 and 418–4472].

Major comments:

  • It is not correct to include the former-drinkers into the non-drinkers.

Response: We agree with your comment; previously, we had performed the analysis per your suggestion, but there were some problems. Therefore, the contents of the manuscript have been revised as follows.

  1. Previously, we analyzed the data of former drinkers separately. There was no statistical significance. Thus, we analyzed former drinkers as non-drinkers and secured statistical significance in the DNA methylation analysis [Lines 453–458].
  2. This KoGES cohort is being tracked and analyzed, and the DNA methylation analysis is in progress with more subjects. In the future DNA methylation analysis targeting more subjects, we will analyze the following three groups: non-drinkers, former drinkers, and drinkers.

  • You have found that, in the male group with non-CVDs (N=183), there is much more drinkers (N=136) than non-drinkers (N=47). Does it mean that the alcohol use in man is a certain protection against CVD?
  • According to the Table 1, drinkers are particularly men with higher education and higher income. Does it mean that the alcohol use is a stress compensation? Then, there is larger environment effect in men, and women have higher genetic predisposition. Can you see differences in genetic or epigenetic predisposition to drinking between male and female?

Response: The relevant information has been added in Section 4.4 (Limitations). The cohort was relatively small, and therefore, there might have been a bias; this should be addressed in future studies [Lines 459–463].

  • In the Discussion, you should include the chapter about SNP results and remove the chapter 4.1 that is redundant (basic information) and you should summarize your results of DNA methylation.

Response: We have revised Section 4.1 and added Section 4.4 related to SNPs.

Minor comments:

  • In the text, you have nonsense sentences (line 29-30, 127-128, 134-140) that should be removed.

Response: The comments inserted in the manuscript during the editing process have been deleted.

  • The chapter “Definition & Measurement of Lifestyle Factors” should be explained in more details and clearly.

Response: Irrelevant contents have been deleted, and explanations for variables and coding methods for exercise have bene added. Similarly, we have added explanations for variables in the legend of Figure 1 [Lines 147–152].

  • In the Table 1, it is not explained meaning Exercise1, Exercise2, Exercise3, Exercise4, and Exercise5.

Response: We have added explanations regarding exercise intensity in Table 2 [Lines 239–246]. Table 1 has been replaced with Table 2.

  • The chapter 3.3 is redundant.

Response: Section 3.3 describes the classification and characteristics of samples and Section 3.4 describes DMR. We have moved the methodology to Section 2.6 [Lines 172 and 182–189], added the explanation in Section 3.2 [lines 212–224], and revised Sections 3.3 and 3.4 to improve clarity [Lines 248–249, 258–263, and 266–267].

  • There are many technical and grammar errors.

Response: The revised manuscript has been edited and rechecked for language and clarity by Editage, a professional English language editing service [WYUCM_312_2]. Please find attached the relevant certificate.

Round 2

Reviewer 1 Report

I do not see a significant change in the manuscript content compared to the first review, I would leave it to the editor to decide.

One final comment: Please make readable figures, the font in most of the figures is too tiny.

Author Response

Response: We have revised our manuscript, and the revised contents were presented in the letter. The resolution of all pictures is high enough, and when enlarged, the font is presented clearly without breaking.

Reviewer 2 Report

You have answered with an elaboration to all of my comments. Moreover, you have revised the manuscript accordingly.

Author Response

Response: The reviewer's meticulous and detailed comments significantly improved the quality of our manuscript. Thank you for your evaluation for the development of this paper.

Reviewer 3 Report

Authors have responded all questions and have reacted to all comments satisfactory.

Author Response

(The authors gave the same response as above.)
